# “*Couch-to-5k* or Couch to Ouch to Couch!?” Who Takes Part in Beginner Runner Programmes in the UK and Is Non-Completion Linked to Musculoskeletal Injury?

**DOI:** 10.3390/ijerph20176682

**Published:** 2023-08-30

**Authors:** Nicola Relph, Sarah L. Taylor, Danielle L. Christian, Paola Dey, Michael B. Owen

**Affiliations:** 1Faculty of Health, Social Work and Medicine, Edge Hill University, Ormskirk, Lancashire L39 4QP, UK; 2Research Institute of Sport and Exercise Science, Liverpool John Moores University, Liverpool L3 3AF, UK; 3Applied Health Research hub (AHRh), University of Central Lancashire, Preston PR1 2HE, UK

**Keywords:** physical activity, exercise, *Couch-to-5k*, musculoskeletal injury, running

## Abstract

Physical activity has mental and physical health benefits; however, globally, three-quarters of the population do not meet physical activity guidelines. The *Couch-to-5k* is a beginner runner programme aimed at increasing physical activity. However, this programme lacks an evidence base, and it is unclear who is attracted to the programme; running also has a high rate of musculoskeletal (MSK) injuries. The aims of this study were to identify the characteristics of people taking part and the incidence of MSK injuries as well as exploring the experiences of people who dropped out of a modified 9-week *Couch-to-5k* programme. A total of 110 runners (average age was 47.1 ± 13.7 years) participated in the study, which involved completion of questionnaires (running experience and footwear information, quality of life (EQ-5D-5L), physical activity level (IPAQ-short form), MSK injury history and knee condition (SNAPPS and KOOS-PS)) at the start, middle and end of the programme and collecting sociodemographic information (age, gender, social economic status, relationship status, education level), as well as body mass index, running experience, footwear information, quality of life, physical activity levels, MSK injuries and knee condition. Fifteen drop-outs were interviewed to explore experiences of the programme. Runners were mainly females (81.8%) with an average age 47.1 years, average body mass index of 28.1 kg.m^2^, mainly from high socio-economic levels, married and educated to degree level. In total, 64% of the sample had previous running experience and were classified as active. Half the sample self-reported pain/discomfort and 37.2% reported anxiety/depression at the start of the programme via the EQ-5D-5L scale. Self-reported health scores increased (*p* = 0.047) between baseline (73.1 ± 18.8 out of 100) and at the midpoint (81.2 ± 11.6), but there were no significant differences between any other time points (end point 79.7 ± 17.5, *p* > 0.05). Twenty-one injuries were reported during the programme (19%). Previous injury increased the risk of new injury (OR 7.56 95% CI from 2.06 to 27.75). Only 27.3% completed the programme. Three themes emerged from interviews; MSK injury, negative emotions linked to non-completion and design of the programme. The *Couch-to-5k* may not attract diverse inactive populations, but future work with larger sample sizes is needed to substantiate this finding. Dropping out was linked to MSK injury and progressive design, so future programmes should consider including injury prevention advice and more flexible designs.

## 1. Introduction

There is strong scientific evidence to support the positive dose-response relationship between physical activity and chronic disease morbidity and mortality, highlighting the importance of physical inactivity as a public health issue [1,2,3,4,5]. The current World Health Organisation (WHO) guidelines state that adults aged 18–64 years should participate in at least 150 min of moderate-intensity aerobic activity or at least 75 min of vigorous-intensity aerobic activity, or an equivalent combination of moderate- and vigorous-intensity activity throughout the week [6]. However, global estimates indicate that only one in four adults are sufficiently active, with only “slow and uneven” progress being made in this area [7,8]. 

Running is one form of physical activity that may help contribute to meeting these guidelines. In England, between 2018 and 2019, 6.6 million adults reported running at least twice in the last 28 days; this increased to 7.3 million during the COVID-19 pandemic and reduced to 6.5 million in 2022 [9]. Running is one of the top three sport and leisure time activities adults participate in [10], and it is beneficial for health as it is associated with a reduction in all-cause mortality [11]. A popular beginner running initiative in the United Kingdom (UK) is the *Couch-to-5k*, invented by recreational runner, Josh Clark, in 1996 [12], which involves running three times a week with progressive increments over a nine-week period [12]. Although the programme can be followed by individuals, physical activity initiatives have increased effectiveness when combined with community-wide campaigns [13], and where exercise is a group norm, physical activity levels may increase due to an increase in group identity [14,15], specifically in running activities [16]. 

Interestingly, there is no empirical evidence to support the design of *Couch-to-5k*, despite its being central to the Office for Health Improvement and Disparities recent campaigns to increase people’s activity levels, which are often promoted through the UK National Health Service (NHS) [17]. Indeed, a UK government press release published in January 2023 championed the *Couch-to-5k* programme, stating that the accompanying app had been downloaded 6.5 million times since it launched in 2016, and users had completed over 6 million runs in 2022 alone [17]. Such health apps have the potential to improve population physical activity levels using embedded behaviour change techniques, such as goal setting [18]. The most popular app is the NHS version; 322,700 people had rated the NHS *Couch-to-5k* app an average of 4.8 stars out of 5, and it was number 25 in all Health and Fitness downloads on the Apple© app store [19]. However, research is needed to identify characteristics and experiences of runners on this programme. 

Furthermore, while the effect of running on the cardiovascular system in inactive groups, such as those targeted by the *Couch-to-5k* is well understood [4,5], there is limited research that describes the impacts on the musculoskeletal (MSK) system [20]. An understanding of MSK injury in novice runners is important as this is a deterrent from future engagement [21]. Smits et al. [22] reported that 78% of those injured were still absent from a running programme after six weeks. Buist et al. [23] stated that 40% of women and 37% of men did not re-start running after injury, and this pattern was higher in novice runners (48%) than those already engaged in running (24%). Evidence of the injury incidence for *Couch-to-5k* runners is limited; one paper reported an incidence of 49% [24], but more research is needed as it is unclear if MSK injury is a reason for non-completion of the *Couch-to-5k* programme and future deterrent of physical activity [20]. 

It is therefore also important to identify the risk factors for MSK injury in novice runners to develop injury prevention programmes. Age may be an influencing factor, but both younger and older runners have been reported as having more injuries [25,26,27]. Higher BMI values (≥30 kg.m^2^) increase the risk of a running injury [23,24,26,27], and a BMI < 20 kg.m^2^ may reduce the risk of an injury [26]. Previous injury (though not necessarily running related) and less running experience also increased injury rates [24,25,26,27]. New footwear was also attributed to a greater risk of injury [27]. The knee is the most common location of running injuries [24]. It is important to identify whether these risk factors are apparent in *Couch-to-5k* runners. 

In summary, it is still unclear who participates in the *Couch-to-5k* programme and if injury incidence is related to non-completion. Therefore, the aims of this study are to:Identify the characteristics of people taking part in a modified *Couch-to-5k* programme.Identify the incidence of MSK injuries and potential risk factors for people taking part in a modified *Couch-to-5k* programme.Explore the experiences of drop-outs of the modified *Couch-to-5k* programme.

The quantitative hypothesis is that MSK injuries will be related to non-completion. 

The qualitative research question is: what are the experiences of people who do not complete the *Couch-to-5k* programme?

## 2. Materials and Methods

### 2.1. Research Design

The study was an observational cohort study using quantitative and qualitative methods and took place between May 2018 and May 2020. Quantitative data was collected using questionnaires at baseline, mid-way through and at the end of the programme. Runners who dropped out of the programme were also sent a questionnaire and asked to be interviewed. The completed interviews comprised the qualitative data. The study was carried out at a sports centre in North West England, UK. 

### 2.2. Participants

A convenience sample was used to recruit 110 participants from North West England to a *Couch-to-5k* programme. A power analysis was not performed; the sample size was determined by the number of participants willing to volunteer during the length of the study. Participants were recruited following attendance at a voluntary presentation session in which the lead researcher introduced and explained the study. The average age was 47.1 ± 13.7 (range 17–75) years (females 46.7 ± 13.6; males 48.9 ± 14.3 years), and 81.8% were female. The inclusion criteria were any adults who registered for the *Couch-to-5k* programme, could provide written informed consent and could communicate in English. 

### 2.3. Procedures

The programme delivery was modified to include both community and individual level social support following the literature [13,14,15,16] (see Table 1). One group session a week was delivered on the running track at a local sports facility by a trained instructor. However, the session content was as per the original programme The two additional runs each week were self-directed through the *Couch-to-5k* NHS app as per the original programme [12]. All participants were asked to adhere to the three runs per week prescribed by the *Couch-to-5k* programme, but this was not systematically monitored. 

### 2.4. Questionnaire Data

Runners were asked to complete a questionnaire booklet during week one, week five and week nine of the programme. Drop-outs were also sent a questionnaire within two weeks of non-attendance. The following variables were collected (see Appendix A for more details): ○Socio-demographics at baseline only (age, gender, social economic status (SES), relationship status, education level) ○Self-reported body mass index (weight (kg)/height (m)^2^),○Running experience using the following questions: Have you ever participated in running recreationally before? If yes, please expand. ○Footwear perceptions using the following five questions:
Did you get your running footwear fitted? Yes/NoIs your running footwear New/Used?How much did your running footwear cost?How would you describe your footwear? Running specific/general sports/fashion.How important do you think running footwear is to injury prevention? Very important/important/not sure/not important/not important at all. ○Quality of Life (QoL) using the EQ-5D-5L [28]. This measurement tool is a generic instrument for describing and valuing health using five dimensions: mobility, self-care, usual activities, pain/discomfort and anxiety/depression. The tool also includes a “Health Today Scale”, in which 0 represents the worst health the participant can imagine and 100 represents the best health the participant can imagine. Cronbach’s alpha is reported as 0.82 and ICC as 0.78 [95% CIs 0.65 to 0.86] [29]. ○Physical activity level using the IPAQ-Short Form [30]. This measurement tool collects information on vigorous and moderate exercise and walking and sitting time. Physical activity levels were then defined as low (total PA MET-minutes per week <600), medium (total PA MET-minutes per week >600 and high (total PA MET-minutes/week >3000). (The corresponding correlation coefficient for test-retest reliability is reported as 0.69) [30].○MSK injury history [31] using a standardized self-report form that asked participants if they had an injury that prevented them from doing usual daily activities (including strains, sprains, bursitis, fractures and other injuries to muscle, tendon, bone, joint or ligament). If yes, then the number of injuries, body part injured (from a check list of possible body parts) and cause of the injury was detailed [31]. ○Knee condition using SNAPPS [32] and KOOS-PS [33] questionnaire. SNAPPS is a self-report questionnaire that identifies people with patellofemoral pain and has a Cohen’s kappa of 0.74 [32]. The KOOS-PS reports knee conditions based on activities of daily life and sports and recreation, and it has a reported Cronbach’s alpha of 0.89 [34].

The questionnaire booklet was distributed by a member of the research team in person and via post for drop-outs. 

### 2.5. Interviews

All runners who completed a drop-out questionnaire (regardless of non-completion reason) were asked to take part in an interview. The semi-structured interviews, lasted between 45 and 90 min and were audio-recorded on a digital voice recorder. The 11 interview questions were based on previous literature in this area and designed to explore personal insight for drop out. These questions explored perceptions of the programme, the impact of the programme on participants’ health and wellbeing, potential injury episodes, and knowledge on injury prevention and treatments (see Appendix B for interview questions). 

### 2.6. Data Analysis

Questionnaire data was imported into SPSS (IBM, Version 25). A descriptive analysis provided information on socio-demographic characteristics of runners at baseline. Pearson’s chi-squared analysis compared differences between categorical data (running experience and physical activity levels). Changes in QoL visual analogue score over time were analysed using Friedman’s ANOVAs due to violation of normality. Injury incidence was presented as a percentage of total runners and number of injuries recorded per runner. To assess potential risk factors, comparisons of age, BMI, previous injury, running experience and trainer condition between injured and non-injured runners were completed using Mann–Whitney U tests (due to violation of normality) and chi-squared analysis. A logistical regression was used to present the odds ratio (OR) of completing the programme based on injury history. Significance was accepted at *p* < 0.05. 

Audio-recordings from the interviews were transcribed verbatim. An inductive and data-driven analytical strategy was used to identify and discuss the salient themes repeated across and within the transcripts [35]. Thematic analysis was utilized as it allowed for the identification of patterns and meaning across a dataset and provided a flexible approach [33]. Inductive thematic analysis of the data [36] was completed by one author (DC). Codes and emergent themes were checked by a second author (MO) to ensure consistency of coding. To ensure methodological rigor, credibility and trustworthiness [37,38], final themes were cross-examined against the data in reverse from the themes to the data sheets. Any disagreements were discussed between authors (DC and MO) until an agreement was reached. The Braun and Clarke “15-point Checklist of Criteria for Good Thematic Analysis” [36] was used to ensure quality in the analysis process. 

## 3. Results

### 3.1. Questionnaire Completion

Figure 1 details questionnaire completions and number of drop-outs. At the mid-point, 48.2% of the baseline sample had dropped out of the programme, and a further 16.4% dropped out by the end point, meaning a total drop out of 64.5% from baseline. Of the total runners who dropped out of the intervention, 74.6% of these were before the mid-way point. Twenty-one (29.6%) of the runners who ceased the programme completed a questionnaire booklet at the time of drop-out; unfortunately, reasons were not provided as to why more individuals did not return the questionnaire at this point. 

### 3.2. Characteristics of People Taking Part in a Modified Delivery Couch-to-5k Programme

Table 2 presents participants’ characteristics. Baseline BMI was 28.1 ± 6.0 kg/m^2^ (overweight) (range 17.9 to 53.8). The largest proportion of runners were classified at lower professional and higher technical occupations (37.4%), followed by higher professional occupations (26.3%) and intermediate occupations (17.2%). Most runners were married (57.3%). The largest proportion of runners were educated up to degree level (61.8%), and a further 27.3% of the sample were educated to A levels or O levels. 

Most of the group were moderately active (43.5%), followed by highly active (30.6%), with only 25.9% classified as having low activity levels at baseline. At baseline, 70 out of the 110 participants (64%) reported having recreational running experience (ꭓ^2^, (2, *n* = 109) = 6.21, *p* = 0.045), and 30 (43%) of them stated that this was from a previous *Couch-to-5k* attempt. Seven people reported that they had previously completed the programme. Other types of running experience mentioned by participants included gym running (*n* = 4), *parkrun* (*n* = 13), charity/fun/organised runs (*n* = 11), running clubs/groups (*n* = 5), 10k distance (*n* = 10) and half marathon distance (*n* = 2). Of those who stated they had running experience, 40 were participating on their own (58.0%), 14 with a friend (20.3%) and 15 with a partner (21.7%). Of those who stated that they did not have running experience, 16 were running on their own (40.0%), 17 with a friend (42.5%) and 7 with a partner (17.5%). 

Most runners had not had their footwear fitted (75.2%) (ꭓ^2^, (2, *n* = 109) = 27.75, *p* < 0.001) and started the programme in used footwear (72.7%) (ꭓ^2^, (2, *n* = 109) = 23.86, *p* < 0.001), with the highest proportion being over 12 months old (64.7%). The average spend was £58.7 ± 27.4 (range £7 to £140). A total of 47.7% of runners wore running-specific shoes whilst the remaining 52.3% wore general/fashion shoes (ꭓ^2^, (2, *n* = 109) = 0.23, *p* > 0.05). Almost 90% thought that running footwear was either important or very important to injury prevention (ꭓ^2^, (2, *n* = 109) = 39.43, *p* < 0.001), significantly more than the not sure or not important responders. 

Most of the runners reported no problems with mobility (89.1%), self-care (98.2%) and usual activities (90.9%) at baseline. However, 50% reported either slight or moderate problems with pain/discomfort, and 37.2% reported either slight or moderate problems with anxiety/depression at baseline. The visual analogue scale (health today scale) score significantly changed over the programme (ꭓ^2^ (2) = 7.27, *p* = 0.03). Dunn–Bonferroni post hoc tests were carried out, and there was an increase (*p* = 0.047) between baseline (73.1 ± 18.8 out of 100) and at the midpoint (81.2 ± 11.6), but were no significant differences between any other time points (end point 79.7 ± 17.5, *p* > 0.05). 

### 3.3. The Incidence of MSK Injuries and Potential Risk Factors of People Taking Part in a Modified Couch-to-5k Programme

Table 3 provides detail on injury data. At baseline, 22 runners (8 male, 14 female) (21.2%) reported having an injury in the past 12 months. Previous injury caused a decrease or stop in physical activity in 87.5% of this group, and most participants (72.7%) sought advice from a health care professional. The most common location of previous injuries was the knee (25%). Most runners (37.5%) stated the cause of previous injury to be “other” reasons that included occupation and “general old age”. Of these 22 participants, 13 (59%) had dropped out of the programme by the mid-way point, and only six completed the programme. 

The SNAPPS survey revealed 17 (15.5%) runners had current knee pain (one had patellofemoral pain), four of whom also reported a previous injury in the last 12 months. Nine of these runners dropped out of the programme. The average score for all participants on the KOOS-PS scale was 2.3 ± 3.5 at baseline, which represents 89.5/100, with 100 being no difficulty with knees [39].

The total injury incidence rate was 19%, or 0.19, per runner and 21 injuries were reported by 20 runners. One runner reported two injures, the first at the mid-way point and then another at their drop out point before the end of the programme. Nine runners reported one injury and had dropped out by the mid-way point, and four runners reported one injury and had dropped out by the end. Six runners completed the programme but reported an injury at the end point. 

This sub-group of runners who reported an injury were not significantly different in age, BMI, running experience, trainer condition (new versus used) or injury history compared to those who were uninjured and completed the programme (all *p* > 0.05). However, regression analysis revealed that previous injury was a significant predictor of whether the runner finished the *Couch-to-5k* programme (OR = 7.56, 95% CIs 2.06 to 27.75), as runners who finished the programme were more likely to not have a history of injury. Furthermore, the KOOS-PS score was significantly worse in the drop out data (10.2 ± 5.8) than the finishers (1.4 ± 2.3) (U = 9.0, *p* < 0.001). 

### 3.4. Experiences of Drop-Outs of the Modified Couch-to-5k Programme

All of the 21 people who were unable to complete the programme were invited to interview; a total of 15 people (11 women, 4 men; age range: 28 to 66 years; socio-economic classification range from higher professional occupations to semi-routine occupations) were interviewed (71%). Inductive analysis of the interviews revealed three main themes: musculoskeletal injury, negative emotions linked to non-completion and the progressive design of the programme. 

### 3.5. Musculoskeletal Injury

A musculoskeletal injury was the reported reason for drop out in 73% (11/15) of participants. Participants stated their surprise at being injured, and some reported trying to ‘run through’ the pain.

“*I did try to work through the pain, but I just couldn’t. It was as soon as I started to run it started hurting again and I thought, uh oh, best not I might be doing more damage.*” (PN13)

Participants reported a fear that the injury might cause permanent damage and were unaware of how the injury had happened in some cases. A general lack of knowledge and understanding of sports injuries and treatment was reported. 

“*It was terrible pain. I couldn’t put any pressure on my knee at all. I couldn’t walk.*” (PN43) 

The injuries were more concerning to the participants when they impacted their daily life and were reported as the main reason for not continuing with the programme. 

### 3.6. Negative Emotions Linked to Non-Completion

Participants reported several negative emotions linked to non-completion of the programme, often due to an injury episode. These included frustration, embarrassment, disappointment, guilt and devastation at having to drop out. 

“*I felt a bit of an idiot, as I say. I felt, how can you fail in the first week? … I thought what an idiot*” (PN9)

“*To be honest I’m absolutely devastated because I was really, really enjoying the sessions*” (PN27)

“*I am frustrated about it. I’m upset about it really because I really wanted to, and I was enjoying it, and I felt healthier.*” (PN36)

These negative emotions were heightened due to the positive experiences and outcomes the participants had felt when starting the programme. Participants initially reported a sense of inclusivity due to coming from similar fitness backgrounds, all having a shared goal and overall social interaction. This illustrated the positive experience participants felt being part of the programme at the start, but when asked about their overall programme experience, participants often reflected on negative experiences. 

“*Erm…I’d say probably the negative because I couldn’t finish it because I just felt like I’d let myself down and you know, the other people that were doing it. Yeah, definitely thinking about it, it was the negatives of not being able to do it and feeling a bit of a failure I suppose.*”(PN128)

Participants indicated feelings of low self-worth as a result of not being able to remain in the programme and missed the social elements. The negative emotions experienced could impact future running participation, with participants noting the intensity of running and potential for injuries as barriers to future engagement.

### 3.7. Progressive Design of the Programme

The programme was seen to increase in physical difficulty quickly, often too quickly for some participants who were new to running, which appeared to contribute to drop-out. 

“*It did get harder. But if it had been over instead of say a 10 week it be a 20 week, you know and it was spread out a bit more, I think it would be a lot easier. But I’m not saying you wouldn’t get injuries. It might for the people who don’t really exercise, it would be a lot easier.*”(PN9)

Some reported an inability to keep up with the programme prescription after the first couple of weeks, when they noticed the intensity increased. This progressive nature of the programme was linked to the timing of injury episodes, with participants noting that the injury episodes occurred alongside the increases in the amount of running required. Participants reported that, if they missed prescribed sessions, they felt unable to keep up with others in the group upon their return and felt uncomfortable or unable to return after missing multiple sessions. 

“*Monday you are doing 5 (minutes running), Wednesday you are doing 8 then Friday you are doing 20! And that is a massive leap.*”(PN9) 

“*I really don’t think it is a Couch-to-5k. I think it’s if you’ve done a bit of running before and want to get back into it, then we’ll help you do it.*”(PN7)

It was noted that participants understood the programme needed progression for them to improve, but the speed of the progression was problematic. This was especially prominent for those who reported that they were new to running. 

## 4. Discussion

The average characteristics of runners attending the first session of a modified *Couch-to-5k* programme were middle-aged (47.1 ± 13.7 years), overweight, married women, educated to degree level and with higher SES. This is a positive finding regarding older age, women and higher BMI, as these are some key characteristics that public health physical activity interventions aim to target. For example, the 2020/2021 census data in England reported that 24.2% [95% CIs 24 to 24.5] of women were inactive (defined as engaging in less than 30 min of physical activity per week), compared to 22.4% [from 22.4 to 22.6] of men [9]. Furthermore, as age and BMI increases, physical activity levels decline [40,41]. However, this sample does include high educational and SES levels, which are not diverse, and people in national statistics socio-economic classification (NS-SEC) 1–2 are more likely to be active [9]. This may be explained as the programme was delivered in a local authority area of which 39.5% of the population work in level 1 and 2 occupations [42]. Unfortunately, to the best of our knowledge, there are no previous studies on the characteristics of *Couch-to-5k* participants, and we are therefore unable to compare our study sample to a “typical” population. Future research should investigate a larger range of SES groups participating in the *Couch-to-5k* to see if the current findings are transferable. 

An interesting finding was that over 70% of the sample self-reported that they were moderate or highly active at baseline and had previous running experience (43% had previously attempted the *Couch-to-5k*). Furthermore, most of the runners who self-reported inactive levels at baseline dropped out of the programme. It is positive that people were happy to re-attempt the programme, but this may also suggest that *Couch-to-5k* may not attract inactive populations, as is intended. Running may have a negative perception among inactive populations, and other activities, such as walking, could perhaps be promoted instead [8,43]. For example, Active 10, the NHS step counting app, is reported to be more acceptable than other forms of physical activity with inactive populations [44]. However, it should also be noted that this finding may have been influenced by study participation error as people who were more active may have been more willing to take part in the study. 

Regarding injury, at baseline, 35 runners (31.2%) reported a history of injury and/or knee pain, and 25 of them dropped out of the programme. The cause of previous injury was mainly reported as “other” and included occupation and “old age”. Benyamini and Burns [45] reported that age is significantly associated with perceived health status and, hence, perception of ability to perform physical activity. At the end of the programme, 21 injuries had been reported by 20 runners, equating to a 19% injury rate, or 0.19 injuries per runner. Runners who were injured were no different in age, BMI, running experience, trainers or injury history compared with non-injured runners at baseline. However, finishers of the programme were more likely to report no injury history and had better KOOS-PS scores than dropouts. Linton and Valentin [24] reported an injury rate of 49.8% of *parkrunners*. However, we are unaware of any comparative injury incidences reported from the *Couch-to-5k*; therefore, future research with a larger sample size should aim to validate this finding. 

The drop-out rate reported in this study (64%) is not unique to this *Couch-to-5K* programme. Stevinson et al. [46] reported adherence as 53 ± 27% from another *Couch-to-5K* programme. *parkrun* events have reported drop-out rates of 37% at 6 months and 46.5% at 12 months following registration [47]. Moreover, 63% of new gym members will abandon activities before the third month, and less than 4% will remain for more than 12 months of continuous activity [48]. This could be due to a multitude of factors, including ongoing social support, motivation, resources and cost, and further exploration is warranted.

Non-completion of the *Couch-to-5k* programme in this instance, either through injury, work or illness, led to negative feelings of frustration, embarrassment, disappointment and guilt. Individuals who sustain serious sport or physical activity injuries often do not return to physical activity either in the short-term [22] or permanently [25,47]. Indeed, participants in the current study were fearful of the longer-term impact of their injury related to work and family and were reluctant to return to physical activity for risk of reinjury. The negative emotions could cause a barrier to future and sustained physical activity engagement. Therefore, organisers and designers of future running programmes for beginners should be aware of the potential long-term impacts of injury on an individual’s wider lifestyle and potential disengagement with physical activity as a result.

Prior to dropping out of the programme, participants initially reported enjoyment, inclusivity, shared goals and a sense of community through the social support provided. Previous beginner running programmes have found similar results. Stevinson, Plateau, Plunkett, Fitzpatrick, Ojo, Moran and Clemes [46] found that adherence to a *Couch-to-5k* programme was positively correlated with social support, enjoyment, motivation, confidence and satisfaction with progress. However, similar to the current study, Stevinson and colleagues [46] also found injury to be the most common reason for missing sessions or discontinuing training. In some cases, injury can impact more than just engagement with physical activity, with Hootman, Macera, Ainsworth, Addy, Martin and Blair [25] stating that 21.1% of men and 13.2% of women missed work after sustaining an injury. This highlights two key aspects to consider for programme designers for population-level physical activity interventions: first, the importance of social support in physical activity engagement, and second, injury prevention to maintain this engagement [49]. Future programmes should pay close attention to loading requirements for beginners [50] and build in social support [51] and injury support, as well as guidance for those who do suffer an injury, encouraging to re-engage with physical activity when they are able.

Drop-outs perceived the *Couch-to-5k* programme to progress too quickly, especially in week 5, where the sustained running time increases from 5 min, to 8 min, to 20 min, and linked this aggressive progression to injury. A supervised training programme for a marathon found injuries increased as load increased [52]. Conversely, previously sedentary, predominately overweight men and women randomized to a gradually-paced 12-month aerobic exercise intervention with a goal of 360 min/wk did not report more injuries or bodily pain at 12 months than the control group [31]. This difference may be due to the more graduated increase in exercise time for the first 10 weeks of the 12-month programme. Furthermore, participants in the current study reported that, if they missed prescribed sessions, they felt unable to keep up with the group upon their return, and felt unable to return after missing multiple sessions as the programme had progressed too much. A focus on individuals’ fitness and injury history at the start programmes may allow for a more tailored and graduated increase in running time and distance, supported by a longer programme overall. This could be advantageous in lowering injury and dropout rates and warrants further exploration in future running programmes.

Considering these findings, it is important that beginner running programmes include injury prevention (for example footwear and dietary advice) and education on preparation for physical activity. There was some evidence that runners were partly aware of prevention methods; a total of 90% believed footwear to be important or very important, but advice on MSK injury prevention and the role of previous injury alongside physical activity recommendations is limited. For example, the current advice from NHS England is the inclusion of a warm-up and cool down before and after exercising, respectively, and completing generic strength and flexibility training, but no further recommendations are provided. Furthermore, the UK chief medical officer guidelines [53] include a small and generic section on muscle and bone strengthening and balance training, but these are not specific to beginners. This advice is not evidence-based, perhaps because is no current evidence on the best-practice injury prevention techniques in beginner exercisers [54]. Furthermore, sport injury prevention literature is heavily biased toward recreational, pre-professional and professional exercisers.

### Strengths and Limitations

A strength of this study is the mixed-method data collected and presented, which provides both descriptive and contextual information relating to engagement in the programme. This data is also an important addition to the current lack of evidence surrounding the *Couch-to-5k* programme. It is, however, important to acknowledge that the current study evaluated a modified *Couch-to-5k* programme with a social support element. It is hoped that further qualitative data collected from the current sample exploring this social support element in more detail will be published. Future research is needed to see if current findings are consistent with people who download and use the traditionally formatted *Couch-to-5k* based app. Further limitations of the study include small sample size and biases within the recruitment strategy and sampling. For example, there was an unequal gender split and lack of diversity in SES, yet it is not clear whether this was due to the location of the study or issues with the reach of the *Couch-to-5k* programme. Given that injury was self-reported, it can be argued that complete injury diagnosis was not achieved as clinical tests conducted by a professional is required to achieve this. However, self-reporting area of injury is justified.

## 5. Conclusions

The *Couch-to-5k* programme has the potential to attract groups who are more likely to be inactive (women, overweight and middle aged). However, a key finding of this study was the high level of dropout during this *Couch-to-5k* programme. This drop out was linked to the programme’s progressive design and suffering potential MSK injury. Therefore, future research should consider how to better support individuals during *Couch-to-5k* and other beginner physical activity initiatives. MSK injury prevention information should be provided at the start of the programme, including strength and flexibility exercise and injury treatment options. Follow-ups with participants who dropout and tailored MSK injury advice would also be useful.

## Figures and Tables

**Figure 1 ijerph-20-06682-f001:**
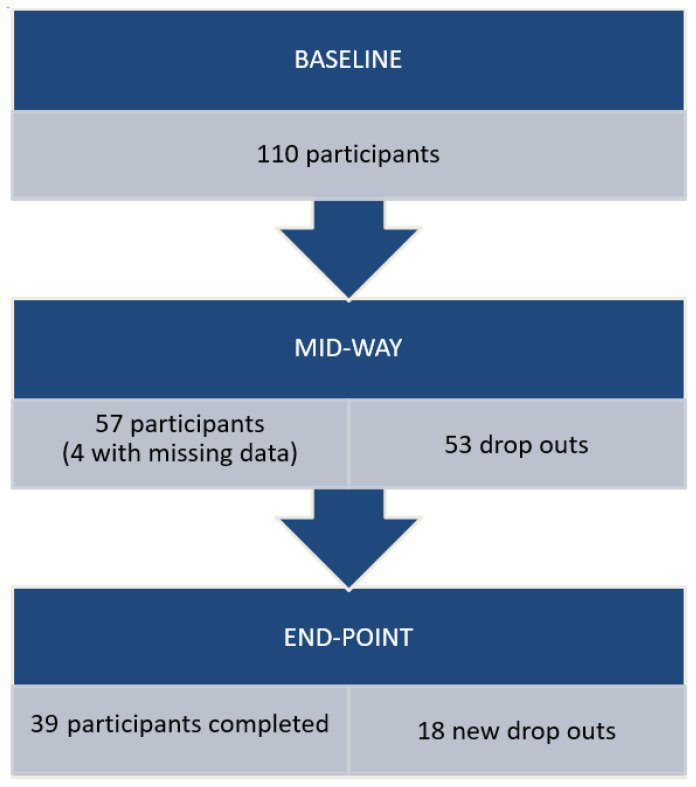
Respondents and drop-outs.

**Table 1 ijerph-20-06682-t001:** The modified delivery of *Couch-to-5k*, including the one instructor-led run each week, based on [12].

	Run 1 (Instructor-Led)	Run 2 (Self-Led)	Run 3 (Self-Led)
Week 1	Brisk 5-min walk, then 1-min run + 1.5-min walk ×8	Brisk 5-min walk, then 1-min run + 1.5-min walk ×8	Brisk 5-min walk, then 1-min run + 1.5-min walk ×8
Week 2	Brisk 5-min walk, then 1.5-min run + 2-min walk ×6	Brisk 5-min walk, then 1.5-min run + 2-min walk ×6	Brisk 5-min walk, then 1.5-min run + 2-min walk ×6
Week 3	Brisk 5-min walk, then 1.5-min run + 1.5-min walk +3-min run + 3-min walk ×2	Brisk 5-min walk, then 1.5-min run + 1.5-min walk +3-min run + 3-min walk ×2	Brisk 5-min walk, then 1.5-min run + 1.5-min walk +3-min run + 3-min walk ×2
Week 4	Brisk 5-min walk, then 3-min run + 1.5-min walk + 5-min run + 2.5-min walk + 3-min run + 1.5-min walk + 5-min run	Brisk 5-min walk, then 3-min run + 1.5-min walk + 5-min run + 2.5-min walk + 3-min run + 1.5-min walk + 5-min run	Brisk 5-min walk, then 3-min run + 1.5-min walk + 5-min run + 2.5-min walk + 3-min run + 1.5-min walk + 5-min run
Week 5	Brisk 5-min walk, then 5-min run + 3-min walk + 5-min run + 3-min walk + 5-min run	Brisk 5-min walk, then 8-min run + 5-min walk + 8-min run	Brisk 5-min walk, then 20-min run
Week 6	Brisk 5-min walk, then 5-min run + 3-min walk + 8-min run + 3-min walk + 5-min run	Brisk 5-min walk, then 10-min run + 3-min walk + 10-min run	Brisk 5-min walk, then 25-min run
Week 7	Brisk 5-min walk, then 25-min run	Brisk 5-min walk, then 25-min run	Brisk 5-min walk, then 25-min run
Week 8	Brisk 5-min walk, then 28-min run	Brisk 5-min walk, then 28-min run	Brisk 5-min walk, then 28-min run
Week 9	Brisk 5-min walk, then 30-min run	Brisk 5-min walk, then 30-min run	Brisk 5-min walk, then 30-min run

**Table 2 ijerph-20-06682-t002:** Runner Characteristics at baseline. BMI = Body Mass Index. SES = Socio-economic status.

Variable	N	Count	%
**Age (years)** <18 18–39 40–59 ≥60	105	1 27 58 19	0.9 24.5 52.7 17.3
**Gender**	110		
Female		90	81.8
Male		20	18.2
**BMI (kg.m^2^)**	94		
Underweight Normal Overweight Obese		2 28 32 32	1.8 25.5 29.1 29.1
**SES Class** 1.0 1.2 2.0 3.0 4.0 5.0 6.0 7.0	99	3 26 37 17 3 3 6 4	2.7 23.6 33.6 15.5 2.7 2.7 5.5 3.6
**Relationship**	109		
Single		34	30.9
Married		63	57.3
Civil partnership		1	0.9
Divorced		7	6.4
Widowed		4	3.7
**Education**	110		
None O levels entry O levels passes A levels Degree Other		2 9 10 20 68 1	1.8 8.2 9.1 18.2 61.8 0.9
**Running Experience**	110		
No		40	36.4
Yes		70	63.6
**Footwear** Used New	109	80 29	72.7 26.4
**Footwear**	109		
Running Specific		52	47.7
General sports		52	47.7
Fashion		5	4.6
**Footwear**	109		
Fitted		27	24.5
Not fitted		82	74.5
**Footwear**			
Average cost (mean ± SD)	102	£58.66 ± 27.4	
**How important is footwear?**			
Very important	109	65	59.1
Important		32	29.1
Not sure		12	10.9
**Quality of Life (No problems)** Mobility Self-care Usual Activities Pain/discomfort Anxiety/Depression	110	98 108 100 55 68	89.1 98.2 90.9 50 61.8
Health score (mean ± SD)		73.75 ± 18.9	
**Physical Activity Levels**			
Low		22	25.9
Moderate	85	37	43.5
High		26	30.6

**Table 3 ijerph-20-06682-t003:** Injury details. Baseline data is from the 12 months prior to the programme, mid data is for the first half of the programme, end data is from the mid-point to the end of the programme and dropouts is injury anytime during the programme.

	Baseline		Mid		End		Dropouts	
	*n*	%	*n*	%	*n*	%	*n*	%
**Injury in the past 12 months or during Couch-to-5k**								
No	82	78.8	45	88.2	34	87.2	9	47.4
Yes	22	21.2	6	11.8	5	12.8	10	52.6
**No. of injuries in the past 12 months or during Couch-to-5k**								
1	16	76.2	4	66.7	4	80.0	9	90.0
2	4	19.0	1	16.7	1	20.0	1	10.0
3	1	4.8	1	16.7	0	0	0	0
**Doctor or health care practitioner**								
No	16	72.7	6	100	3	40.0	4	40.0
Yes	6	27.3	0	0	2	60.0	6	60.0
**Injury location**								
Hip	1	4.2	0	0	0	0	1	8.3
Groin	0	0	0	0	1	25.0	1	8.3
Upper Leg	2	8.3	0	0	0	0	1	8.3
Knee	6	25.0	2	33.3	0	0	3	25.0
Lower leg	1	4.2	3	50	2	50.0	2	16.6
Ankle	3	12.5	0	0	0	0	2	16.6
Foot	2	8.3	0	0	1	25.0	0	0
Other	9	37.5	1	16.6	0	0	2	16.6
**Side of the body**								
Left	3	13.0	2	33.3	4	80.0	4	44.4
Right	14	60.9	2	33.3	0	0	2	22.2
Both Sides	6	26.1	2	33.3	1	20.0	3	33.3
**Injury Cause**								
Occupation	1	4.2	0	0	0	0	0	0
Leisure	3	12.5	0	0	0	0	3	30.0
Home Maintenance	1	4.2	0	0	0	0	1	10.0
Sports/PA	7	29.2	6	100	4	80.0	6	60.0
Transport	3	12.5	0	0	0	0	0	0
Other	9	37.5	0	0	1	20.0	0	0
**Changes to PA in response to injury**								
No	3	12.5	1	16.7	3	60.0	0	0
Yes, decrease	14	58.3	4	66.7	2	40.0	3	30.0
Yes, stop	7	29.2	1	16.7	0	0	7	70.0
Yes, increase	0	0	0	0	0	0	0	0
**Injury Treatment**								
Home Treatment	11	27.5	6	100	4	57.1	5	35.7
X-Ray/MRI/CT	7	17.5	0	0	0	0	1	7.1
Medication	7	17.5	0	0	2	28.6	2	14.3
Physical Therapy	13	32.5	0	0	1	14.3	6	42.9
Surgery	2	5	0	0	0	0	0	0

## Data Availability

Research data is available on request from the corresponding author.

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
