# Peer review of "Couch-to-5k or Couch to Ouch to Couch!?” Who Takes Part in Beginner Runner Programmes in the UK and Is Non-Completion Linked to Musculoskeletal Injury?"

_ijerph, 2023, doi:10.3390/ijerph20176682_

Round 1

Reviewer 1 Report

Running, in its various forms, is currently regarded as an easily approachable, affordable and widely accepted first approach to physical exercise and overall active lifestyle, owing to its aerobical nature, synergy with adequate dietary patterns and social angle. Multiple programmes aimed at sedentary lifestyles have included running as an invitation to physical activity, often with a "zero-to-hero" approach.
In this paper, the Authors examine a sample cohort of one such programme, Couch-to-5k, detailing its demographics, salient characteristics as well as the reasons for programme drop-out and the role of muscoloskeletal injuries in the matter.
I find the Authors' work to be useful indeed in assessing overall characteristics of the programme's users, as well as their Health Related Quality of Life: such data might be invaluable in further refining this tool as well as coping with the issues a naive runner might encounter.

I have a few considerations on the paper:

- Only 29,6% of dropouts completed questionnaires: were any reasons given, or they simply did not mail the questionnaires back ?
- Footwear is of prime importance in running: an incorrect fitting (size, foot type, etc.), especially in patients with previous injuries, leads to severe pain, imbalance and injury. Dedicated running shoes' age and mileage are equally important, as they correlate with degradation of the midsole, often strongly adding to the risk of injury. I noticed that only footwear age was analyzed between the two groups: I would suggest delving deeper into footwear appropriateness, if possible, or taking the matter into account in future studies.
- Results stemming from physical activity are often diminished if not negated altogether due to persistent dietary mistakes. In such a programme, it could very well cause lack of improvement, which in turn ties into inability to cope with the increased effort required, drop-out and subsequent effects in health-related quality of life. Why weren't dietary patterns investigated?
- Re-injury is a rather crucial matter: the Authors' tied previous injuries to greater odds of not finishing the programme, which is rather understandable as the text seems to hint at chronic diseases by themselves responsible for dminished physical performance. As the Authors indeed noticed, professional appraisal of injury type and severity as well as the assessment of incorrect habits (e.g. "powering through" with sustained exercise routines or overloading) might prove useful. 

Minor points

- In the paper's introduction, the Authors state " It is important to identify if these risk factors are related to Couch-to-5k runners", and yet the study has no control group. It would've been interesting to compare results with a cohort of subjects undergoing a similar programme with full-time professional guidance. As it stands however, the study design does not allow any such assumptions to be made. 
- SES is used in the abstract, yet it would be preferable to avoid undeclared acronyms
- I would suggest further elucidating baseline physical activeness assessment via IPAQ-SF.
 - Table 3: the caption unclearly explains the data presented, e.g. how most data points except the first refer to the "previous injuries" subgroup. As an added example, the "injuries in the last 12 months" is unclear as in which data is presented for the mid-point and end-programme milestones: I assume new ones arisen during the study?

Author Response

Please see file attached. 

Reviewer 2 Report

Reviewer Comment

Although it is interesting in terms of the subject of the study, there are many places in the form of submission of the manuscript that do not meet the article writing criteria. The methodological framework is not fully explained in the summary. There are deficiencies in the data of individuals. The questionnaires used are unknown. Not all data are disclosed in the results. The entire summary should be thoroughly reorganized to include the method of the study and its important consequences. In the introduction, there are some places that are written without showing the source. References should be added to these sentences. The purpose should be presented in a more understandable way. In the articles, there are hypotheses in the studies in the quantitative design, and research questions in the studies in the qualitative design. There are no hypotheses and research questions at the end of the introduction. After the purpose of the study is stated, the hypothesis and research question should be added. There are deficiencies in all sub-headings in the method section. The study design is not clear, there are missing data on the participants. The evaluation questionnaires used were not disclosed. All these deficiencies must be corrected. P values should be added to the tables in the results. Abbreviations and statistical methods used should be explained under the table. The data obtained with the qualitative pattern could have been presented better. These data should be reviewed again. The discussion should have been more systematic and the conclusions should not contain personal comments. All inferences should be supported by the literature source. It should not be forgotten that the form and quality of the presentation are as valuable as the work itself. You should present your work in a much better way. Extensive editing is required throughout the manuscript. Please make all suggestions carefully and in an orderly manner.

Revision

1.      The title of the study does not fully reflect the research. The title and summary should be written more interestingly and reflective of the content as it is the place where the first information about the study is taken for the readers. Change the part of the title that is a multi-method cohort study and revise your title with a more comprehensive one.

2.      It is not specified which measurement questionnaire was used to evaluate the evaluations made in the method section of the summary. These should be explained. In addition, the age range and average age information of the participants should be added. How did you determine the presence of anxiety and depression that you mentioned in the conclusion part of the summary? The diagnosis of depression was determined by the feedback of the individuals or by the diagnosis. In addition, the comparison of the data of the 3 evaluation periods and the statistical significance value are not given in the summary. The explanations you make in the abstract do not adequately reflect the manuscript, please make the necessary adjustments.

3.      “However, there are over 10 versions of 5k running applications aimed at beginners. It appears Couch-to-5k is popular despite its lack of empirical evidence base, and research is needed to identify characteristics and experiences of runners on this programme.” You must cite all the information you use in the introduction. None of the information you provide without reference is suitable for article writing. It is better to give your personal comment at the end of the Introduction when you state the purpose of the study. Make the necessary arrangements. Add source reference.

4.      “While the effect of running on the cardiovascular system in inactive groups such as those targeted by the Couch-to-5k is well understood[4,5], there is limited research that describes the impacts on the musculoskeletal (MSK) system.” In the second sentence, you mentioned that there are a limited number of studies, but you did not give a clue as to which of these studies were. Include the sources of these studies at the end of this sentence.

5.      “however, more research is needed as it is unclear if MSK injury is a reason for non-completion of the Couch-to-5k programme and future deterrent of physical activity.” You should cite the source that the causes of musculoskeletal injuries are not clear.

6.      At the end of the introduction, the purpose of the research should be stated more simply. Hypotheses for quantitative data, research questions for qualitative data should be created and added at the end of the introduction.

7.      “2.1. Research Design”

This section is very simple, short and incompletely explained. In this section, the short plan of the study should be explained. Your study is a mixed study planned in both quantitative and qualitative patterns. This must be specified. It should be briefly explained what will be evaluated at which stages. In addition, information should be added where the study was carried out, in which center, the evaluations were made by experts who are experts in which field and how many years experienced in this field. Include where the ethics committee permission was obtained, the ethics committee number and date information in this section. Those who read this section should be able to understand the methodological structure of the study. Make the necessary arrangements.

8.      “2.2. Participants”

It should be explained in more detail in the Participants section. The information in which age range individuals are included should be added. How was the number of participants determined? Has power analysis been done? If it was done, it should be explained in this section with reference to what was done and with which program. The inclusion and exclusion criteria should also be explained in this section. The flowchart of the participants should be attached here. Include the source reference of the “Couch-to-5k” program. How did you determine the number of evaluated participants in the qualitative design? Make the necessary arrangements.

9.      “2.4. Questionnaire Data”

Quality of Life (QoL) (EQ-5D-5L[27])

o Physical activity Level (IPAQ-Short Form[28])

o MSK injury history[29]

o Knee condition (SNAPPS[30] and KOOS-PS[31]) These questionnaires should be explained in detail. How many questions does it consist of? How many headings and subheadings are there? Created by whom, what are the reliability coefficients? How are the answers? How is the scoring? What does an increase or decrease in score mean? All this should be explained. The information about how long it takes in total to complete all these survey evaluations should also be added.

10.  “2.3. Interviews”

How were semi-structured interview questions created? What did you refer to when creating these questions? How many questions does it consist of? What device was the audio recorded with? How long did the interviews last on average? How many people were interviewed? Add this information.

11.  If you could create sub-themes under the main themes you created for the data in the presentation in the qualitative design, the reasons could be revealed better, and the data definitely contains this information. Examine the data again to form sub-themes. In this way, those who will be much more systematic and understandable.

12.  Move the flowchart in Figure 1 to the participants section?

13.  Write the explanations of BMI and SES abbreviations under Table 2

14.  “Aim 1. Identify the characteristics of people taking part in a modified Couch-to-5k programme.”  It is not correct to express this title as a purpose. “Aim 1.” ‘delete. Edit the title like this: “1. Characteristics of people participating in a modified Couch-to-5k program” Arrange Objectives 2 and 3 accordingly.

15.  In Table 3, add the time-dependent statistical significance (p value) value in the comparison between the data of 3 periods. In addition, if post-hoc comparison tables are created, the effects can be observed better. Make the necessary arrangements.

16.  The name of the statistical analysis used should be added at the bottom of the table.

17.  Quantitative evaluations in the method section are not given in the result section. These data results should be given in tables.

18.  “The aims of this study were to (1) identify the characteristics of people taking part in a modified Couch-to-5k programme, (2) identify the incidence of MSK injuries and poten-tial risk factors, and (3) explore the experiences of drop-outs.” There is no need for such an explanation in the discussion section, delete it.

19.  “The average characteristics of runners attending the first session of a modified Couch-to-5k programme were middle-aged (range 17-75 years),” It is a contradictory statement that you say that the average characteristics of individuals are middle-aged and show the age range of 17-75. Either express the age range as that of the middle age group or edit your expression.

Author Response

Please see file attached

Reviewer 3 Report

My recommendations are the following:

Abstract - The first two sentences in the abstract, I recommend to reformulate, make an estimate not supported by the bibliographic index. Attention to punctuation: ex. Three themes emerged from interviews; MSK swear, possibly two points. I recommend that you detail what SES represents.

I recommend replacing the keyword public health, too general, not relevant

Extension of sections 2.1.

In section 2.2. I recommend entering part of the data presented in table 2.

In the abstract you mention that the program was carried out for 10 weeks and according to table 1 only 9, I recommend clarification.

I recommend mentioning the α-Cronhbach index for each applied questionnaire.

In the first part of the Discussion, the previously mentioned ideas and results are repeated, the second paragraph, I recommend clarification.

I recommend reorganizing and expanding the Discussions section, with making comparisons with data from previous studies. You focus too much on the previously mentioned results.

I recommend that the Conclusions section be rewritten, focused on the recorded results and not on recommendations.

In conclusion, the idea is interesting, but the method of organization is cumbersome, the results are mentioned in blocks and difficult to follow.

You adapted your program and it is observed that the incidence of accidents is very high, looking at the details of the program, it is observed that in weeks 7-9 the running time is between 25-30 after 5 minutes of walking. It would be interesting to detail the abandonment and the incidence of accidents by age category. Because the sample has a very wide range between 17-over 70 years. I would like to highlight the fact that the program was not adapted by age category, only adapted, which is also reflected in the low results as benefits.

Starting from the statement that the population is inactive, in this sense if such adapted programs are implemented, then probably the active ones will become inactive in the future.

I believe that the implemented program has a design error from the beginning.

Author Response

Please see file attached

Reviewer 4 Report

In their evaluation of the Couch-to-5k program, the authors prevent timely and useful insights in the participation, non-completion and injury incidence by program participants. Mixed methods are used, comprising of surveys of participants and interviews with program drop-outs. Although the study has merit and underpins a program that supports important public health goals, I have some concerns with the manuscript in its current form.

Major concerns

There is insufficient information provided in the Methods section for the reader to understand how program participants were selected into the research and if/how this resulted in bias. It is described as: “A convenience sample was used to recruit 110 participants from the North-West of England to a modified Couch-to-5k programme. The inclusion criteria were any adults who registered onto the Couch-to-5k programme, could provide written informed consent and communicate in English”. How was the program advertised and how did study participants opt-in to the research? Please describe (also giving dates of recruitment period). Most importantly, how does the research study sample compare to the program participants overall? In general, women and older persons are reported to have higher study participation rates (i.e. statistically demonstrate a greater willingness to participate in research). Could you quantify any possible biases in your sample and explain how this may have affected your results?

The study sample is quite small and I suspect that some of your analyses are underpowered. Can you provide a power calculation for the logistic regression analysis please? Small study sample is not addressed in the limitations section (I think it should be) although it is mentioned in the first sentence of the conclusion.

Minor concerns

The manuscript needs carefully proofreading with particular attention to use of punctuation.

In the abstract, ‘progressive design’ is mentioned. This term becomes clear after reading the paper but the abstract should stand on its own – therefore, please explain further or remove this term.

Please revise the term ‘trainers’ in the introduction. Does this refer to footwear?

Methods section, Questionnaire data: BMI is not a sociodemographic variable.

Appendix 1 is referred to in the methods; however, there is no appendix.

How were activity levels defined (low; moderate; highly active)? Please provide this information in the Methods section.

The results of the Health Today Scale are given: “Dunn-Bonferroni post hoc tests were carried out and there was an increase (p = 0.047) between baseline (73.1±18.8 out of 100) and at the midpoint (81.2±11.6) but were no significant differences between any other time points (end point 79.7±17.5, p>0.05).” Could you please explain how to interpret this, for readers not familiar with the Health Today Scale?

The information provided in Table 3 is confusing, i.e. ‘…injuries in the past 12 months (baseline)/ since starting Couch-to-5k’. Are these new injuries, pre-existing injuries or a mix? In the context, I would expect this table to describe new injuries only. The injury cause categories in Table 3 are also confusing. Are new running injuries included under ‘Sports/PA’? Why not have a category for injuries that occurred from the program? In Table 3, diagnostic imaging is included under treatment (which it isn’t).

In the discussion (second paragraph) - please place this in the context of potential study participation bias.

In the discussion perhaps you can touch on the issue of program drop-out vs. study drop-out. Perhaps adherence to the Couch-to-5k program was improved because of participation in the research study - or it may have had the opposite effect. 

Author Response

Please see file attached. 

Round 2

Reviewer 2 Report

Reviewer Comment

I think the Manuscript is in better shape than the first version with fixes. Although many of the corrections given have been made, there are still areas that need to be corrected. The end of the introduction should be expressed more concisely. The transition between paragraphs and sentences in the introduction should not be sharp. Transitions should be more connected. The part of the method section, especially on qualitative design work, should be more descriptive and a checklist should be used. Results should be expressed more clearly and tables should contain all information. The comments in the discussion section contain a lot of personal opinions. these should be corrected. Comments should be made by citing the literature. In general, a more fluent and academic language should be used throughout the manuscript. Please make the suggested edits completely.

1.      Carry the information between the dates of the study in the participants section to the research design section.

2.      Information on how the number of participants was determined, whether power analysis was performed, and if so, how it was done should be added to the Participants section.

3.      Use a checklist for the method in the qualitative pattern.

4.      Information on how the qualitative questions are prepared and how many questions the question pool consists of should be added to the method section.

5.      Add the p values you specified in the temporal comparison to table 3. Also, add the information about which statistical method you used at the bottom of the table.

6.      The statement that you stated in a precise way at the beginning of the discussion that there is no work done in this area is a very clear statement. Always consider the possibility of your research and not being able to reach you. In this context, such an expression; Please state that according to the research we have done, we have not been able to reach such a research.

7.      The comments and inferences made in the Discussion section contained too many personal comments. These comments should be made by citing sources from the literature. Make the necessary arrangements.

Reviewer 3 Report

No comments

Author Response

Many thanks for your review of this work. 

Reviewer 4 Report

Thank you for addressing the concerns raised in the review process.

Author Response

(The authors gave the same response as above.)
